# Thinking globally to improve care locally: A Delphi study protocol to achieve international clinical consensus on best-practice end-of-life communication with adolescents and young adults with cancer

Ursula M. Sansom-Daly[1,2,3]*, Lori Wiener[4], Anne-Sophie Darlington[5], Hanneke Poort[6], Abby R. Rosenberg[7,8,9], Meaghann S. Weaver[10,11], Fiona Schulte[12,13], Antoinette Anazodo[2,3,14], Celeste Phillips[15], Louise Sue[16], Anthony R. Herbert[17,18], Jennifer W. Mack[19], Toni Lindsay[20], Holly Evans[1,2], Claire E. Wakefield[1,2], on behalf of *The Global Adolescent and Young Adult Cancer Accord End-of-Life Study Group*¶

1 Behavioural Sciences Unit proudly supported by the Kids with Cancer Foundation, Kids Cancer Centre, Sydney Children's Hospital, Randwick, New South Wales, Australia, 2 School of Clinical Medicine, UNSW Medicine & Health, Randwick Clinical Campus, Discipline of Paediatrics, UNSW Sydney, Kensington, New South Wales, Australia, 3 Sydney Youth Cancer Service, Prince of Wales/Sydney Children's Hospital, Randwick, New South Wales, Australia, 4 Psychosocial Support and Research Program, Pediatric Oncology Branch, Center for Cancer Research, National Cancer Institute, National Institutes of Health, Bethesda, Maryland, United States of America, 5 School of Health Sciences, University of Southampton, Highfield, Southampton, United Kingdom, 6 Department of Psychosocial Oncology and Palliative Care, Dana-Farber Cancer Institute, Boston, Massachusetts, United States of America, 7 Division of Hematology/Oncology, Department of Pediatrics, University of Washington School of Medicine, Seattle, Washington, United States of America, 8 Palliative Care and Resilience Program, Seattle Children's Research Institute, Seattle, Washington, United States of America, 9 Cambia Palliative Care Center of Excellence, University of Washington School of Medicine, Seattle, Washington, United States of America, 10 Divisions of Palliative Care & Pediatric Hematology/Oncology, University of Nebraska Medical Center, Lincoln, Nebraska, United States of America, 11 National Center for Ethics in Health Care, Washington DC, Washington, United States of America, 12 Division of Psychosocial Oncology, Department of Oncology, Cumming School of Medicine, University of Calgary, Calgary, Alberta, Canada, 13 Hematology, Oncology and Transplant Program, Alberta Children's Hospital, Calgary, Alberta, Canada, 14 Kids Cancer Centre, Sydney Children's Hospital, Randwick, New South Wales, Australia, 15 School of Nursing, Indiana University, Indianapolis, Indiana, United States of America, 16 Adolescent and Young Adult Cancer Services Team, Canterbury District Health Board, Christchurch, New Zealand, 17 Children's Health Queensland Hospital and Health Service, South Brisbane, Queensland, Australia, 18 Centre for Children's Research, Queensland University of Technology, Brisbane, Queensland, Australia, 19 Population Sciences for Pediatric Hematology/Oncology, Dana-Farber Cancer Institute, Boston, Massachusetts, United States of America, 20 Chris O'Brien Lifehouse Cancer Centre, Camperdown, New South Wales, Australia

¶ Membership of The Global Adolescent and Young Adult Cancer Accord End-of-Life Study Group is provided in the Acknowledgments

* ursula@unsw.edu.au

## Abstract

For the sizeable subset of adolescents and young adults whose cancer is incurable, developmentally appropriate end-of-life discussions are critical. Standards of care for adolescent and young adult end-of-life communication have been established, however, many health-professionals do not feel confident leading these conversations, leaving gaps in the implementation of best-practice end-of-life communication. We present a protocol for a Delphi

**Data Availability Statement:** Deidentified research data will be made publicly available when the study is completed and published.

**Funding:** Ursula Sansom-Daly received an AYA Global Accord Psycho-Oncology Research Acceleration Grant numbered RG180972 from the Adolescent and Young Adult Cancer Global Accord (a partnership between Teen Cancer America [https://teencanceramerica.org/], Teenage Cancer Trust [https://www.teenagecancertrust.org/] and Canteen [https://www.canteen.org.au/]). The funders will not have a role in study design, data collection and analysis, decision to publish or preparation of the manuscript. Ursula Sansom-Daly is also supported by an Early Career Fellowship from the Cancer Institute of New South Wales (ID: 2020/ECF1163) and an Early Career Fellowship from the National Health and Medical Research Council of Australia (APP2008300). Lori Wiener is supported, in part, by the Intramural Program of the National Cancer Institutes, Center for Cancer Research. Dr. Rosenberg has received grants for unrelated work from the National Institutes of Health, the American Cancer Society, Arthur Vining Davis Foundations, Cambia Health Solutions, Conquer Cancer Foundation of ASCO, CureSearch for Children's Cancer, the National Palliative Care Research Center, and the Seattle Children's Research Institute. The opinions herein represent those of the authors and not necessarily those of their institutions or funders. Claire Wakefield is supported by a Career Development Fellowship from the National Health and Medical Research Council of Australia (APP2008300). The Behavioural Sciences Unit is proudly supported by the Kids with Cancer Foundation, by the Kids Cancer Alliance, as well as a Cancer Council New South Wales Program Grant (PG16-02) with the support of the Estate of the Late Harry McPaul. There was no additional external funding received for this study.

**Competing interests:** The authors have declared that no competing interests exist.

study informing the development and implementation of clinician training to strengthen health-professionals' capacity in end-of-life conversations. Our approach will inform training to address barriers to end-of-life communication with adolescents and young adults across Westernized *Adolescent and Young Adult Cancer Global Accord* countries. The *Adolescent and Young Adult Cancer Global Accord* team involves 26 investigators from Australia, New Zealand, the United States, Canada and the United Kingdom. Twenty-four consumers, including adolescents and young adults with cancer history and carers, informed study design. We describe methodology for a modified Delphi questionnaire. The questionnaire aims to determine optimal timing for end-of-life communication with adolescents and young adults, practice-related content needed in clinician training for end-of-life communication with adolescents and young adults, and desireability of evidence-based training models. Round 1 involves an expert panel of investigators identifying appropriate questionnaire items. Rounds 2 and 3 involve questionnaires of international multidisciplinary health-professionals, followed by further input by adolescents and young adults. A second stage of research will design health-professional training to support best-practice end-of-life communication. The outcomes of this iterative and participatory research will directly inform the implementation of best-practice end-of-life communication across *Adolescent and Young Adult Cancer Global Accord* countries. Barriers and training preferences identified will directly contribute to developing clinician-training resources. Our results will provide a framework to support further investigating end-of-life communication with adolescents and young adults across diverse countries. Our experiences also highlight effective methodology in undertaking highly collaborative global research.

## Introduction

As the world becomes increasingly global, so too do our patient cohorts and the healthcare systems that serve them. Advancing the field of adolescent and young adult (AYA) oncology/haematology research and practice for patients aged 15–39 [1] requires an increasingly inclusive approach to ensure culturally-informed perspectives [2,3]. Despite considerable gains in recent decades, numerous research-to-practice implementation gaps in end-of-life communication best-practice remain, in part driven by the scarcity of trained AYA-focused health-professionals [4,5]. Collaborative, global approaches can bridge these gaps by harnessing the highly specialized, finite and dispersed population of AYA oncology/haematology health-professionals to match the needs of an increasingly diverse, global workforce [6], and the AYAs they care for [2,3]. As the field of AYA oncology/haematology evolves, some patients may also undertake international travel while pursuing access to further or different treatments not available in their home country [7,8]. Advancing science and practice in AYA oncology/haematology therefore requires global research approaches, and an examination of how standards of care may be implemented in different countries.

Psychologically, a cancer diagnosis confronts young people with their mortality whilst they are busy planning their futures. The fact that these futures may be prematurely cut short is unique to the context of AYAs, and thus highly distressing and developmentally challenging [9,10]. Further, AYA cancer mortality remains, on average, higher than pediatric cancer mortality [11–13]. Given these unique challenges, ensuring best-practice end-of-life care and communication for AYAs remains a clinical imperative in this era of novel therapies, precision

medicines, and uncertain prognoses [14]. Due to medical advances across Western and economically developed nations, 80–88% of AYAs now survive their disease [13,15–17]. However, 12–20% of AYAs in Westernized countries die within 5 years of diagnosis, [13,16,17] due in part to the predominance of diagnostic delays, lack of access to appropriate care (including recruitment onto clinical trials) [4,18], plus the incidence of rare, treatment-resistant disease [15,19]. In low and middle-income countries, the proportion of AYAs who die is even greater [20]. In 2019 this led to ~396,000 AYAs aged 15–39 dying from cancer worldwide [21]. End-of-life considerations are critical for this group. Yet reviews have indicated that currently, most of the literature exploring end-of-life communication and care among AYAs has come from the United States (US) [22]. This limits the capacity of the field to improve end-of-life communication practices that will benefit a broader, more diverse population of AYAs across countries.

We define end-of-life communication as conversations about death and dying, preferences for medical and psychosocial care towards end of life, including issues of prognosis when cure is not likely, as well as broader quality of life, social, psychological and existential concerns patients may have in the context of a life-limiting illness [22]. These conversations can be between the dying young person and their treating healthcare professionals, families and friends. These conversations often occur in the last 12 months of an individual's life and/or when a clinical team *"would not be surprised if the patient died in the next 12 months"* [23]. Other clinical guidelines highlight the relevance of these conversations whenever an individual is living with a potentially life-limiting and fatal disease, regardless of the length or uncertainty of the trajectory/future prognosis [24–26]. The population of AYAs for whom these conversations are likely to be relevant and important is also rapidly expanding. With the evolution of early-phase clinical trials, modern oncology/haematology treatments are increasingly blurring the boundaries between receiving active treatment with some curative intent and extending life. The advent of precision medicine and targeted therapies is also transforming some cancers into chronic illnesses, managed or stabilized using novel therapies with an uncertain long-term prognosis. The expansion of this prognostic 'grey zone' that exists between curable and incurable disease necessitates new approaches to end-of-life communication with AYAs.

Best-practice guidance from Westernized countries suggests that AYAs should be actively involved in these kinds of end-of-life conversations, together with their parents/caregivers, to the extent that they would like to be [22,26]. Comprehensive, age-appropriate end-of-life communication with AYAs can therefore shift the clinical focus from simply avoiding potential negative outcomes to also exploring and affirming potential positive, meaningful aspects of life for AYAs with advanced cancer and their families [27]. Providing AYAs with the opportunity to engage in end-of-life conversations supports their ability to live well prior to dying. Optimal end-of-life communication can facilitate early intervention which can lead to symptom and supportive care interventions that promote good quality of life rather than emphasise intensive medically focussed interventions. This is true across cancer types including haematological diagnoses [28,29]. For example, compared with AYAs who do have the opportunity to engage in end-of-life conversations, AYAs who are not provided the opportunity to talk about end-of-life issues may die in a state of 'emotional isolation', with greater anxiety, and poorly managed pain [9,30–32]. They may also be less likely to die at their location of choice (e.g., at home) [33], and more likely to experience intrusive interventions in the days and weeks before their death [30,31,34].

Parents and family members may also be at a greater risk of developing persistent complex bereavement, complicated grief, regret, anxiety, and depression if not supported to speak with their AYA family member about end-of-life issues and preferences [35,36]. Literature from Western cultures suggests that families expect the healthcare team to take the lead in

introducing end-of-life topics [22]. Consequently, even when end-of-life conversations are relevant, without specific intervention or steps taken by the healthcare team, these conversations occur in <3% of AYAs/families [37–39]. This phenomenon of families waiting for healthcare professionals to signal the 'right time' to start these discussions may also be compounded by the poor shared understanding that exists between AYAs and their parents/carers regarding when the AYA patient wants to have end-of-life conversations; in one study from the US, while 86% of adolescents wanted 'early timing' of end-of-life conversations, only 39% of their families knew this [40]. Therefore, the capacity of the health-professionals to guide and facilitate age-appropriate end-of-life conversations for AYAs is critical.

International work by our team has established best-practice standards of care for end-of-life communication [25,26,41]. Based on a rigorous synthesis of the existing evidence, these standards make the recommendation that: *"Youth with cancer and their families should be introduced to palliative care concepts to reduce suffering throughout the disease process, regardless of disease status,"* and *"When necessary youth and families should receive developmentally appropriate end-of-life care"* [26]. However, early evidence across several countries (including the US) indicates that this standard is likely not implemented well across centres [25,41,42]. Little is known about whether and how this standard of care is implemented beyond the US. Some data highlight that even if end-of-life conversations do occur with AYAs, they often happen too late [43].

Numerous barriers may contribute to this gap, at individual, healthcare-team, and hospital-system levels [22]. Recent work has highlighted barriers at a young person and familial level [44,45]. At the individual health professional level, clinicians such as doctors, nurses and allied health professionals can experience considerable distress and counter-transference when health-professionals (who may be relatively young themselves) treat and support dying young people [46,47]. Clinicians can also feel uncomfortable discussing end-of-life issues, and may avoid doing so to avoid feelings that they are 'failing' or have 'let down' their patient [48], or their worry that such conversations diminish hope [49]. Team and system level factors can also complicate the delivery of best-practice end-of-life communication. Lack of staffing capacity and specialist training, and some inter-disciplinary clinicians' beliefs around the role, potential benefits or unique value of palliative care can prevent teams from effectively delivering timely end-of-life communication [25,41].

Training is urgently needed to bridge these practice gaps. Several evidence-based formats have been developed in adult palliative oncology, yet there is little evidence to guide which clinician-training model may be most viable to implement, or translate into the greatest AYA and family benefits [50]. Additionally, the psychological, developmental, health- and family-systems complexities of the AYA years suggests that simply transporting communication training models from adult oncology may be insufficient [51–54].

There are unique aspects to AYA cancer that require additional professional skills and guidance around self-care for health-professionals. However, reflecting the emerging nature of the field, training programs tailored to AYA palliative care are in their infancy [51], with no program currently available internationally to improve clinicians' skills and confidence in end-of-life communication for AYAs with cancer [51]. The type of training needed or preferred by the multidisciplinary workforce of AYA clinicians remains unknown.

In this paper, we describe the methodology for our planned Delphi study of the *AYA Cancer Global Accord* countries (Australia, New Zealand, the United Kingdom, the United States, and Canada) [Stage 1]. We then outline our planned Stage 2 research, which will build on our international Delphi study to develop evidence-based training to support health-professionals in facilitating best-practice end-of-life communication with AYAs. Our overall objective is to develop procedures and methodologies to support an extensive international collaboration to

rigorously study end-of-life communication across several contexts. This model can then be further built upon to examine end-of-life communication across other countries more globally, including non-Westernized nations.

## Methods

### Study context and team

The concept for this study originated at a pre-conference psychosocial research workshop held in Atlanta, US in 2017 prior to the *2nd Global Adolescent and Young Adult Cancer Congress*. A core team originated the study concept during this event (USD, LW, HP, ASD, CP). The investigator team was broadened to include other research and clinical leaders with a focus on end-of-life communication with AYAs, culminating in a final team of 26 international investigators to form the *AYA Cancer Global Accord End-of-Life Study Group*. Our team is highly multidisciplinary, and includes experts trained in the following areas; psychology/clinical psychology ($n = 7$), social work ($n = 2$), oncology/hematology ($n = 5$), nursing ($n = 5$), palliative care ($n = 7$), public health and cancer epidemiology ($n = 3$), psychiatry ($n = 1$), pediatrics ($n = 1$), education ($n = 1$), and clinical ethics ($n = 3$). We represent a collaboration across Australia (10 investigators), New Zealand (1 investigator), USA (10 investigators), Canada (3 investigators), and the UK (2 investigators) and have a combined experience of over 275 years in AYA oncology/haematology, either in a purely clinical, clinical-research, or academic research capacity. We also recruited 24 consumers to inform the final study design. These consumers included AYA survivors ($n = 22$; aged 20–35 years; 84% female), a parent of an AYA survivor ($n = 1$) and a bereaved parent of an AYA who died from their cancer ($n = 1$).

### Aims

Given the unique challenges associated with AYAs communication around end-of-life, this study will focus on the AYA years. Our study builds on the research on evidence-based standards of care and barriers to implementing the standards established by members of our team [25,26,41]. These standards of care exist to guide clinicians in the implementation of best-practice end-of-life communication, however data triangulated across settings suggests that these standards of care may not be being implemented effectively. To ensure all AYAs are able to access this standard of care, we need to understand several things further. Firstly, we need to operationalise best-practice end-of-life communication, for example understanding optimal timing of end-of-life communication with AYAs. We also need to know what training, including format and content, may be helpful to ensure health-professionals are able to implement this operationalised standard.

Therefore, we will use two stages of research to extend this work internationally by establishing multidisciplinary clinical consensus regarding firstly, optimal timing of end-of-life communication; secondly, health-professionals' training needs to support best-practice end-of-life communication; and thirdly, the content and format of health-professional training arising from these data. We aim to answer the following research questions using survey methodologies such as ranking and value weighting:

1. What is the optimal timing for end-of-life communication with AYAs? *(Stage 1).*

2. What practice-related content is needed in clinician training for end-of-life communication with AYAs? *(Stage 1).*

3. How desirable are different evidence-based training models for different health-professionals? *(Stage 1).*

4. What format and content should be included in end-of-life communication training for AYA health-professionals, across disciplines and countries? *(Stage 2)*.

## STAGE 1: Understanding health-professional training needs

**Design.** Adopting a collaborative international approach, this research will use a three-round modified Delphi questionnaire methodology. Delphi methodology is a gold-standard method for establishing consensus across a diverse expert panel [55–57]. Delphi methods attempt to identify answers as close as possible to the 'truth' by minimising bias and balancing perspectives, regardless of stakeholders' status by avoiding real-time confrontation between experts. The method involves presenting participants with an aggregated summary of the first-round questionnaire data and asking them to re-rate their response given the perspectives of their expert peers. This can occur across several iterations to achieve consensus over time. Delphi questionnaires typically involve at least two rounds of data collection from the expert participants [55,56]; our Delphi questionnaire involved three rounds, outlined below.

Round 1 of this questionnaire constitutes a modification to the standard Delphi methodology, by using a separate panel of experts to identify the questionnaire items to be used in the second Round. In Round 2, international participants will be sent the first questionnaire. Participants will be re-contacted approximately 8-weeks following the first round to complete questionnaire items again for Round 3 (Fig 1). As this is an exploratory study in an area where no Delphi questionnaires have been conducted with which to compare our results, we will describe consensus rather than using pre-defined consensus criteria [57].

**Delphi rounds.** *Round 1*: *Expert consultation among study investigators*. This step has already been completed. The expert team of 26 investigators met via teleconference on multiple occasions to discuss the content of both questionnaire rounds of the Delphi. Content was based on literature review and expert investigator opinion and crafted further through iterative discussions. The Principal Investigator (USD) coordinated this process, by organizing two global investigator meetings in May and October of 2019 with 9/26 (34.6%) and 10/26 (38.5%) investigators attending each respectively. Fourteen (54%) investigators were able to attend either meeting. The two meetings advanced the work being developed in a consecutive/iterative way, such that the second meeting built on the first. To supplement these teleconferences, the team also engaged in considerable further discussion and collaboration via email (Table 1); this enabled investigators who were unable to attend either or both meetings to remain involved in discussions as they progressed. After the questionnaire items were finalized across meetings and email correspondence, the questionnaire was piloted by all investigators and further feedback given.

*Round 2*: *Initial Delphi questionnaire—Questionnaire 1*. Questionnaire 1 has been developed based on the Delphi Round 1 expert consultation. The 42-item questionnaire is estimated to take 15–20 minutes to complete based on pilot-testing. Investigators and external collaborators were asked to complete the questionnaire and give feedback with regards to content as well as readability and flow. It will be administered via Qualtrics online questionnaire software. The questionnaire starts with demographic and workplace questions, including questions about religion. The main section covers questions on the most important content for training, preferred training modalities, timing of refresher training, barriers to participating in training, and questions on the optimal timing of end-of-life conversations with AYAs. Questions regarding optimal timing of end-of-life conversations will attempt to take into account various prognostic stages and illness scenarios. For example, participants will be asked when a particular end-of-life topic is appropriate to introduce to AYAs in different illness scenarios.

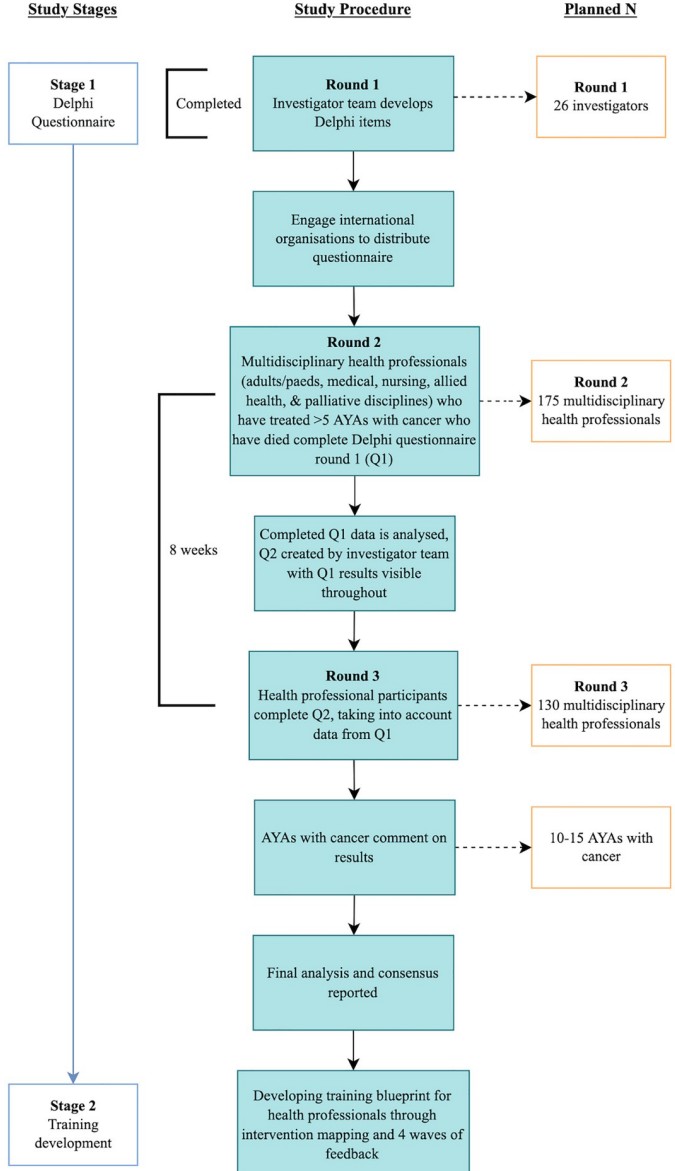

**Fig 1. Flow chart to illustrate Delphi methodology process.**

Additional questions related to participants' individual, clinical experiences in end-of-life communication with different groups (e.g., culturally and linguistically-diverse AYAs, and AYAs with complex psychosocial backgrounds) are also included, as well as questions regarding assessing readiness of AYAs to have these conversations. These will not be part of the repeated, multi-round Delphi portion as these questions relate to health-professionals' individual clinical experiences and are therefore not necessary to generate group-level consensus. These data will instead be used to describe the variability and range of experiences in what clinicians find challenging about end-of-life communication with AYAs. Thus, Questionnaire 1 will include these individual clinical experience questions but Questionnaire 2 will not.

*Round 3*: *Repeat Delphi questionnaire—Questionnaire 2.* The second questionnaire is estimated to take 10–15 minutes to complete and contains 36 questions. This questionnaire has

**Table 1. Counts of group-level emails exchanged between two or more investigators by project topics as at June 2021.**

| Email topics | Emails exchanged |
|---|---|
| Abstracts and presentation input | 62 |
| Regarding AYA (consumer) involvement | 15 |
| From investigators regarding grant updates | 145 |
| Grant logistics (e.g., progress reports, finances) | 116 |
| Organizing investigator meetings | 50 |
| Questionnaire development | 54 |
| Questionnaire distribution | 133 |
| Planning manuscript writing and data analysis | 41 |
| **Total** | **672** |

been developed, together with the Round 1 questionnaire, but not yet administered. The questionnaire will again be administered via Qualtrics, and per Delphi methods contain the same content as Questionnaire 1. This will be overlaid with graphical summaries of the results from Questionnaire 1. For example, question 1 of Questionnaire 2 will ask the same question as question 1 of Questionnaire 1, but will include a graphical summary of results from Questionnaire 1.

*Input from AYAs with a lived experience of cancer.* In addition to including the 24 consumers (including AYAs and parents) who helped shape the study, we will recruit up to an additional 15 AYAs aged 15–40 years who have, or have had, cancer from Australia, the UK and US to be involved in the study's data analysis and interpretation stages (see Delphi questionnaire rounds, below). AYAs with varying cancer diagnoses, across any point of the treatment stage will be eligible to be involved to provide input from a variety of AYA perspectives, however we will not exclusively target recruiting AYAs approaching end-of-life to avoid over-burdening a highly vulnerable group. These AYAs will be recruited through a variety of methods; using social media and community organizations as well as directly via the study investigators. Due to the intervening time period between study design and later data interpretation, as well as a desire not to burden the same individuals, we will conduct a new round of recruitment for these consumers, however consumers originally involved in the study design will be welcome to participate again. These AYAs will be asked to provide feedback on the results of the health-professional Delphi questionnaire by completing an online questionnaire. They will also be invited to attend a teleconference focus group to further discuss results and generate concepts for incorporation into the final analysis, interpretation and publication of the study's results.

## Participants

We will target multidisciplinary health-professionals of varying experience, working with AYAs in either pediatric, AYA-specific, or adult cancer settings internationally. Table 2 depicts our proposed approach and planned sample size. To ensure that all participants have reasonable practice-based knowledge of the area, we defined eligible 'experts' as health-professionals who have provided clinical care to at least five AYAs with cancer who subsequently died. The study has been approved by the South Eastern Sydney Local Health District Human Ethics Committee, and online written consent will be obtained.

We will use multiple methods to recruit expert health-professional participants. Our planned recruitment methods include >67 different avenues such as approaching 30 professional organizations/networks, online/social media, and conference presentations, as well as snowball recruitment via our 26 international investigators (see Table 3). This diverse, multi-

**Table 2. Sampling matrix for Round 2 (Questionnaire 1) by discipline and region; planned sample.**

|  | Australia | New Zealand | Canada | US | UK | Europe | Totals |
|---|---|---|---|---|---|---|---|
| Oncology/hematology: | 7 | 3 | 10 | 10 | 10 | 10 | 50 |
| Nurses: | 7 | 3 | 10 | 10 | 10 | 10 | 50 |
| Palliative care consultants: | 8 | 2 | 10 | 10 | 10 | 10 | 50 |
| Allied health professionals (e.g. psychologists, social workers, play therapists): | 3 | 2 | 5 | 5 | 5 | 5 | 25 |
| AYAs | 3 | 2 | 2 | 3 | 3 | 2 | 15 |
| **Totals** | 28 | 12 | 37 | 38 | 38 | 37 | 190 |

pronged recruitment approach will maximize the reach of the study, a critical approach when trying to capture the experiences of busy health-professionals across settings. This recruitment approach however will mean that it will not be possible to determine true response rates, or differences in our sample of participants relative to study non-respondents or decliners.

## Analysis

Data will be collected using Qualtrics, and then managed with IBM SPSS Statistics Version 27. Data will be stored in files on secure university servers. Central tendencies (means) from Questionnaire 1 will be calculated and presented graphically to Round 2 participants (Questionnaire 2). Final data analysis will occur when Questionnaire 2 is closed and will consist of descriptive analysis to describe the sample demographics, as well as to describe the level of consensus reached for each item. Specifically, means, standard deviations, ranges and modes will

**Table 3. List of professional organizations/networks to be used for Round 2 participant recruitment.**

| | Organization | |
|---|---|---|
| **Australia** | Clinical Oncology Society of Australia | Cancer Nurses Society Australia |
| | APS Psychologists in Oncology Interest Group | Youth Cancer Services |
| | Psycho-Oncology Co-operative Research Group | Eastern Palliative Care Association |
| | Quality of Care Collaborative of Australia (For education in paediatric palliative care) | |
| **Australia/ New Zealand** | Australian and New Zealand Society of Palliative Medicine | Paediatric Palliative Care Australia and New Zealand |
| | Australian and New Zealand Children's Haematology/Oncology Group | AYA Cancer Network Aotearoa |
| **Canada** | Canadian Society of Palliative Care Physicians | Canadian Association of Psychosocial Oncology |
| | Canadian Virtual Hospice | Canadian Partnership Against Cancer AYA National Network |
| **Europe** | European Network for Teenagers and Young Adults with Cancer | European Associate for Palliative Care—Paediatrics Special Interest Group |
| | European Society of Oncology Nursing | |
| **United Kingdom** | Teenagers and Young Adults with Cancer (part of Children's Cancer and Leukaemia Group, UK) | United Kingdom Oncology Nurses Society |
| | Teenage Cancer Trust | |
| **United States** | American Academy of Hospice and Palliative Medicine Pediatrics Special Interest Group | American Society of Pediatric Hematology & Oncology Palliative Care Working Group |
| | Association of Pediatric Oncology Social Workers | American Psycho-Oncology Society End-of-Life and AYA Special Interest Groups |
| | American Academy of Pediatrics Listserv | |
| **International** | Oncology News | International Psycho-Oncology Society (including Pediatrics Special Interest Group) |
| | Pediatric Psycho-Oncology Professionals/Providers International | The International Society of Paediatric Oncology |
| | International Children's Palliative Care Network | |

be used to describe the level of consensus. As such, due to the exploratory nature of the study and of previous literature [57], we will document the degree of consensus and also variability that naturally emerges among our clinician sample in Round 1, then across Round 2 we will use the commonly used threshold of 80% item agreement to determine formal consensus in the final analysis of study results. Qualitative analysis techniques including content analysis will be used to analyze comments from participants, such as additional topic suggestions.

## AYA (consumer) input

We will establish an AYA Advisory Panel comprising an additional 10–15 AYAs aged 15 to 39 to provide feedback on the training priorities identified through our Delphi questionnaire. A further advisory panel of approximately 5 parents of AYA cancer patients/survivors and bereaved parents will be run separately. Following the analysis of results from the Delphi questionnaire Rounds 2 and 3, we will present a written lay summary of key findings to AYAs and parents recruited as consumers. They will be asked to complete an online questionnaire which asks them to reflect on the summary findings with reference to their own cancer-related experiences, record the extent to which the recommendations for health-professional training matches their experience, and identify gaps in the findings. Next, AYAs and parents will be invited to participate in an online focus group, during which the themes of the findings and their reflections on these will be further explored. Finally, they will be invited to share reflections in the publication of the final results. They will be asked to describe ways in which they would like to see AYA and parent input integrated into the development, and potentially the delivery and implementation, of future training programs. This input will not be used to alter the outcomes of the Delphi questionnaire, but rather will contribute a vital layer of qualitative data that will be integral to the interpretation of the data through a consumer-driven lens. The involvement of AYAs and parents as partners in shaping research, particularly AYA health research, is an ethical imperative [58], and can contribute to ensuring that research needed and valued by AYAs translates well to real-world practice [59,60].

## STAGE 2: Training blueprint development and clinical impacts

Following on from Stage 1 and our Delphi questionnaire, we will develop a model for how to train multidisciplinary clinicians to deliver optimal end-of-life communication with AYAs with cancer, including a clear template for what modality, content, and design will be acceptable, feasible, and lead to high uptake [50,51,61]. We have created iterative development opportunities to evolve this plan as the study progresses.

**Intervention mapping.** We will undertake an international scoping exercise to identify and map [62] existing end-of-life communication training, palliative care communication training, oncology/haematology communication training, and serious illness communication training or related programs in either the pediatric or adult sectors. This will ensure that the new training content we develop to address the gaps identified in our Stage 1 questionnaire will build upon existing training offerings internationally. We will also gather data around how existing training programs are delivered and disseminated, and to whom they are accessible (e.g., what health-professional disciplines, in what regions/countries). Few programs have been developed to date targeting this specific skillset among health-professionals working with AYAs with cancer [51]; and many resources may only be available to professionals within certain regions or networks. Data gained from this step will lay a foundation for the later development of our training implementation plan, and will form the basis of developing a flexible, multi-pronged implementation strategy to maximize the number of multidisciplinary health-professionals we are able to reach with our tailored training resources.

**Training development.** Our training development will include undertaking several waves of feedback targeted at refining our training model and its content (below), culminating in a workshop held at an international conference (held in-person or virtually) in the field of psycho-oncology, and an online webinar to disseminate the core pilot content (refined in the workshop) to a broader global audience. Our investigator team will also explore additional pathways where this training might be integrated; this could include AYA-specific oncology/haematology and palliative care training programs internationally [51].

Our pilot training model will be developed and refined through four waves of feedback.

1. AYA feedback: The AYAs recruited internationally as part of our AYA Advisory Panel will assess whether our proposed training addresses the issues and gaps in care they identify as most important, through a combination of online questionnaire and interviews/focus groups.

2. Health-professional feedback: Health-professionals will complete an online questionnaire to evaluate the acceptability, feasibility, and appropriateness of content/design aspects of our training model. We will also use these questionnaires to gain detailed feedback on how best to deliver important AYA-specific content (e.g., psychosocial/cultural issues relevant to end-of-life and the maintenance of the clinical relationship).

3. Interactive workshop: The health-professional workshop will enable a more in-depth, nuanced evaluation of our proposed training model and content with international AYA health-professionals. During the workshop, we will pilot content/exercises (as relevant) and assess how it may need to be tailored to address different cultural needs. At least two investigators will deliver this workshop, either in-person or virtually, together with at least 1–2 AYAs. Acceptability and feasibility (e.g. relevance, intended use) will be assessed immediately afterwards together with health-professionals' characteristics.

4. Online webinar: Having further refined the content of our proposed training through the workshop, we will host an online webinar to further refine and pilot-test this content with a broad health-professional audience. We will offer this pilot webinar as a professional development resource, at no/low cost. This webinar will be targeted toward multi-disciplinary clinicians and promoted through the international networks in our study team. We will record the webinar to enable widespread access. Again, we will evaluate acceptability, feasibility, and relevance of the pilot training model with attendees to inform its future delivery and formal evaluation.

## Discussion

### Progress, challenges, and proposed solutions

Since the commencement of this project, our team via The University of New South Wales has completed three separate ethical (Institutional Review Board) application processes internationally, one governance application, and three protocol amendments for updates to questionnaire content. This required a considerable collaborative effort to coordinate large numbers of documents for three countries' ethics systems. As noted above, most of the collaboration required to develop project materials and review documents has occurred through email (Table 1). The organization of meetings presented logistical challenges in arranging a time suitable for all investigators with respect to nine timezones and varying clinical loads/commitments. To supplement this, 13 comprehensive investigator email updates were also sent between collaborative meetings.

To ensure the global reach required across *AYA Cancer Global Accord* countries (Australia, New Zealand, the United Kingdom, the United States, and Canada), significant effort will be required to promote the effective dissemination and uptake of Round 2 of the Delphi (Questionnaire 1). The team has developed a multi-pronged recruitment strategy, and to date has approached 30 professional organizations across six countries (with six of those organizations being international in nature) who have agreed to disseminate the questionnaire. This process has so far involved the preparation and team review of seven dissemination approval forms. Individual investigators also plan to actively promote global recruitment through their individual professional networks, using snowballing methods to disseminate the questionnaire amongst their local networks, including posting on *listservs* and writing newsletter pieces for local organizations. We have also written an original commentary to launch our study on an international oncology-specific news website [51], which gained considerable attention in its first week (becoming the top-most 'clicked on' article for the website). We have also promoted the study through our social media channels where possible, reaching an audience of >13,200 people internationally as at February 2021. We are hopeful that despite the recent disruption caused by the global Coronavirus (COVID-19) pandemic, this combination of strategies will result in strong recruitment to our study that is appropriately representative across disciplines and countries and will be able to bridge important gaps in the global literature.

## Future directions

**Training evaluation.** Once developed and pilot-tested, we plan to deliver our training to a larger, diverse, multidisciplinary sample of AYA health-professionals across *AYA Global Accord* countries to assess its impact. These evaluations will need to assess both short- and long-term outcomes. In the short-term, following participation in our training model, it will be important to assess AYA health-professionals' self-reported confidence, self-efficacy, and knowledge of different end-of-life communication approaches to navigate different, challenging end-of-life communication scenarios. In the longer-term, we will assess self-reported uptake and use of communication strategies taught during the training model, as well as end-of-life communication experiences. Different members of our team may also seek funding to undertake further satellite research studies building out from this initial work to examine the impact of disseminating this training within different local contexts, from multiple stakeholder- and systems perspectives.

**Training implementation.** Building on all the preceding training content development work, as well as the initial intervention mapping [62] exercise to identify potential existing similar training resources and avenues for training dissemination, we will develop a multi-pronged training implementation strategy to expand access to our new training model. We anticipate that this strategy will be guided by the *Theoretical Domains Framework* [63,64], to account for factors influencing behavior change at individual, team and healthcare system levels, together with the *Consolidated Framework for Implementation Research* [65], to enable further examination of barriers and enablers at different levels of healthcare organizations [66]. This implementation strategy development will be led through our team, in conjunction with *AYA Cancer Global Accord* leaders, and other stakeholders with expertise in end-of-life communication education. We anticipate this implementation plan will involve multiple avenues for implementing this training, and will likely include train-the-trainer components [51,67]. We will look to integrate our new training resources where possible with existing palliative care/end-of-life communication training programs to leverage international engagement and optimize the reach and accessibility of end-of-life communication training to as diverse an international health-professional sample as possible.

## Limitations

While we are encouraged by the potential future impact of our international collaboration, several limitations to the methodology of this research study are worth acknowledging. We are taking an intentionally broad and far-reaching approach with our various recruitment methods, however, our methodology will still rely on data from health-professionals who choose to participate. This may skew our final sample in terms of ultimately including more individuals who may have more experience, or who may already feel more confident, in facilitating conversations about end-of-life topics with AYAs. Our study methodology, including our recruitment approach will also limit our data to Westernized, democratic and mostly English-speaking countries. This pragmatic decision was taken in order to start addressing the gap in the literature first across *AYA Global Accord Alliance* countries, within the constraints of limited funding and the specific remit of this particular funding scheme. This will limit the international generalisability of our findings, given well-acknowledged cultural differences that exist in terms of AYAs', families' and health-professionals attitudes towards end-of-life and palliative care communication [3,68]. This phenomenon, whereby research participants from 'WEIRD' societies (that is, those from Westernized, Educated, Industrialized, Rich and Democratic countries) are over-represented in research is not unique to oncology/haematology [69]. Yet, given ample evidence that standards of care for end-of-life communication are not equitably and consistently implemented even in Westernized countries, this initial step is important to develop the field in an interactive, rigorous way. Future research efforts and implementation strategies will be required to build on the results we obtain here and extend these to different settings, as well as to examine their applicability to more diverse populations within each of our settings (such as First Nations and culturally- and linguistically-diverse groups). Finally, we recognize the under-representation of investigators specialising in education and implementation science on our team, and are in the process of strengthening our team's capacity in this area as the project progresses.

## Conclusions and next steps

The research-to-practice gap in end-of-life communication represents a gap in implementation [70]; developing evidence-based, tailored clinician training is just one important strategy in bridging this gap. Through this collaborative international effort, we are learning from the experiences and perspectives a large group of multidisciplinary health-professionals with experience working with AYAs who have died from cancer, as well as AYAs with lived experience of cancer. Our international approach will contribute to the development of training resources to support multidisciplinary healthcare professionals in enhancing their skills and confidence to lead end-of-life conversations across different healthcare contexts. Important next steps will include validating these training priorities through input by our AYA Advisory Panel, as well as developing a comprehensive training implementation plan.

More broadly, the clinical consensus gained from this research will directly inform what we understand best-practice end-of-life communication to look like in reality, a critical advancement on the published standards to date [25,26,41]. Our results will also be able to contribute to a greater understanding of effective methodology in undertaking highly collaborative international research on this scale. These learnings will be valuable both to individual researchers and to funding agencies wishing to support such collaborative efforts in the future. Further, grounded in evidence-based frameworks of developmentally-appropriate AYA psychosocial care [53], and new social-science paradigms regarding how optimal clinical communication functions to support patient-clinician relationships over time, [71] this research will advance scientific knowledge and end-of-life communication practices with AYAs with cancer across

*AYA Global Accord* countries, to more closely reflect the care they prefer and deserve at end of life.

## Acknowledgments

In addition to the named authors, *The Global Adolescent and Young Adult Cancer Accord End-of-Life Study Group also* included:

Abby Rosenberg; Palliative Care and Resilience Program, Seattle Children's Research Institute, Seattle, Washington, United States of America; Cancer and Blood Disorders Center, Seattle Children's Hospital, Seattle, Washington, United States of America.

Afaf Girgis; Centre for Oncology Education and Research Translation (CONCERT), Ingham Institute for Applied Medical Research, UNSW Sydney, Liverpool, New South Wales, Australia.

Ahmed Al-Awamer; Department of Family and Community Medicine, University of Toronto, Toronto, Ontario, Canada; Princess Margaret Cancer Center, Toronto, Ontario, Canada.

Anne Kirchhoff; University of Utah School of Medicine, Department of Pediatrics, Division of Hematology/Oncology, Salt Lake City, Utah, United States of America; Huntsman Cancer Institute, Cancer Control & Population Sciences, Salt Lake City, Utah, United States of America.

Anne-Sophie Darlington; School of Health Sciences, University of Southampton, Highfield, Southampton, United Kingdom.

Anthony Herbert; Children's Health Queensland Hospital and Health Service, South Brisbane, Queensland, Australia; Centre for Children's Research, Queensland University of Technology, Brisbane, Queensland, Australia.

Antoinette Anazodo; School of Clinical Medicine, UNSW Medicine & Health, Randwick Clinical Campus, Discipline of Paediatrics UNSW Sydney, Kensington, New South Wales, Australia;

Sydney Youth Cancer Service, Prince of Wales/Sydney Children's Hospital, Randwick, New South Wales, Australia.

Celeste Phillips; School of Nursing, Indiana University, Bloomington, Indiana, United States of America.

Claire Wakefield; Behavioural Sciences Unit proudly supported by the Kids with Cancer Foundation, Kids Cancer Centre, Sydney Children's Hospital, Randwick, New South Wales, Australia; School of Clinical Medicine, UNSW Medicine & Health, Randwick Clinical Campus, Discipline of Paediatrics, UNSW Sydney, Kensington, New South Wales, Australia.

Douglas Fair; University of Utah School of Medicine Department of Pediatrics, Division of Pediatric Hematology/Oncology, Salt Lake City, Utah, United States of America.

Fiona Schulte; Division of Psychosocial Oncology, Department of Oncology, Cumming School of Medicine, University of Calgary, Calgary, Alberta, Canada; Hematology, Oncology and Transplant Program, Alberta Children's Hospital, Calgary, Alberta, Canada.

Hanneke Poort; Department of Psychosocial Oncology and Palliative Care, Dana-Farber Cancer Institute, Boston, Massachusetts, United States of America.

Holly Evans; Behavioural Sciences Unit proudly supported by the Kids with Cancer Foundation, Kids Cancer Centre, Sydney Children's Hospital, Randwick, New South Wales, Australia; School of Clinical Medicine, UNSW Medicine & Health, Randwick Clinical Campus, Discipline of Paediatrics, UNSW Sydney, Kensington, New South Wales, Australia.

Jennifer Mack; Population Sciences for Pediatric Hematology/Oncology, Dana-Farber Cancer Institute, Boston, Massachusetts, United States of America.

Joan Haase; School of Nursing, Indiana University, Bloomington, Indiana, United States of America; The RESPECT Signature Center at IUPUI, Indianapolis, Indiana, United States of America.

Karen Wernli; Kaiser Permanente Washington Health Research Institute, Oakland, California, United States of America.

Leigh Donovan; Collaboraide, Queensland, Australia.

Lori Wiener; Psychosocial Support and Research Program, Pediatric Oncology Branch, Center for Cancer Research, National Cancer Institute, National Institutes of Health, Bethesda, Maryland, United States of America.

Louise Sue; Adolescent and Young Adult Cancer Services Team, Canterbury District Health Board, Christchurch, New Zealand.

Maria Cable; School of Nursing, Midwifery and Health, Coventry University, Coventry, United Kingdom.

Meaghann Weaver; Divisions of Palliative Care & Pediatric Hematology/Oncology, University of Nebraska Medical Center, Lincoln, Nebraska, United States of America; National Center for Ethics in Health Care, Washington DC, Washington, United States of America.

Pamela Mosher; Hospital for Sick Children, Toronto, Ontario, Canada.

Richard Cohn; Behavioural Sciences Unit proudly supported by the Kids with Cancer Foundation, Kids Cancer Centre, Sydney Children's Hospital, Randwick, New South Wales, Australia; School of Clinical Medicine, UNSW Medicine & Health, Randwick Clinical Campus, Discipline of Paediatrics, UNSW Sydney, Kensington, New South Wales, Australia;

Ruwanthie (Amanda) Fernando; Palliative Care Service, Liverpool Cancer Therapy Centre, Liverpool Hospital, Liverpool, New South Wales, Australia.

Susan Trethewie; Sydney Children's Hospital, Randwick, New South Wales, Australia.

Toni Lindsay; Chris O'Brien Lifehouse Cancer Centre, Camperdown, New South Wales, Australia.

Ursula Sansom-Daly; Behavioural Sciences Unit proudly supported by the Kids with Cancer Foundation, Kids Cancer Centre, Sydney Children's Hospital, Randwick, New South Wales, Australia; School of Clinical Medicine, UNSW Medicine & Health, Randwick Clinical Campus, Discipline of Paediatrics, UNSW Sydney, Kensington, New South Wales, Australia.

The lead author for this group is Ursula Sansom-Daly, contactable on ursula@unsw.edu.au. Ursula Sansom-Daly wishes to thank the *AYA Cancer Global Accord* mentors who provided helpful guidance in the development and undertaking of this study, in particular A/Prof Pandora Patterson and Dr Fiona McDonald (Canteen Australia), Prof Dan Stark (University of Leeds, UK) and Dr Norma D'Agostino (Princess Margaret Cancer Centre, Canada), and Prof Bradley Zebrack (University of Michigan, USA). The opinions herein represent those of the authors and not necessarily those of their institutions or funders. Meaghann Weaver participated in this project in a private capacity; the views expressed in this article are those of the authors and do not necessarily reflect the position or policy of the U.S. Department of Veterans Affairs, the U.S. Government, or the VA National Center for Ethics in Health Care.

## Author Contributions

**Conceptualization:** Ursula M. Sansom-Daly, Lori Wiener, Anne-Sophie Darlington, Hanneke Poort, Abby R. Rosenberg, Meaghann S. Weaver, Fiona Schulte, Antoinette Anazodo, Celeste Phillips, Louise Sue, Anthony R. Herbert, Jennifer W. Mack, Toni Lindsay, Claire E. Wakefield.

**Data curation:** Holly Evans.

**Formal analysis:** Ursula M. Sansom-Daly, Holly Evans.

**Funding acquisition:** Ursula M. Sansom-Daly, Lori Wiener, Anne-Sophie Darlington, Hanneke Poort, Abby R. Rosenberg, Meaghann S. Weaver, Fiona Schulte, Antoinette Anazodo, Celeste Phillips, Louise Sue, Anthony R. Herbert, Jennifer W. Mack, Toni Lindsay, Claire E. Wakefield.

**Investigation:** Ursula M. Sansom-Daly, Lori Wiener, Anne-Sophie Darlington, Hanneke Poort, Abby R. Rosenberg, Meaghann S. Weaver, Fiona Schulte, Antoinette Anazodo, Celeste Phillips, Louise Sue, Anthony R. Herbert, Jennifer W. Mack, Toni Lindsay, Claire E. Wakefield.

**Methodology:** Ursula M. Sansom-Daly, Lori Wiener, Anne-Sophie Darlington, Abby R. Rosenberg, Celeste Phillips, Louise Sue, Anthony R. Herbert, Jennifer W. Mack, Toni Lindsay, Claire E. Wakefield.

**Project administration:** Ursula M. Sansom-Daly, Holly Evans.

**Resources:** Ursula M. Sansom-Daly.

**Supervision:** Ursula M. Sansom-Daly, Lori Wiener, Anne-Sophie Darlington, Abby R. Rosenberg, Meaghann S. Weaver, Claire E. Wakefield.

**Writing – original draft:** Ursula M. Sansom-Daly, Holly Evans.

**Writing – review & editing:** Ursula M. Sansom-Daly, Lori Wiener, Anne-Sophie Darlington, Hanneke Poort, Abby R. Rosenberg, Meaghann S. Weaver, Fiona Schulte, Antoinette Anazodo, Celeste Phillips, Louise Sue, Anthony R. Herbert, Jennifer W. Mack, Toni Lindsay, Holly Evans, Claire E. Wakefield.

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
