## [Decision Letter · Decision Letter 0]

26 May 2022

PONE-D-22-03862Thinking globally to improve care locally: A Delphi study protocol to achieve international clinical consensus on best-practice end-of-life communication with adolescents and young adults with cancerPLOS ONE

Dear Dr. Sansom-Daly,

Thank you for submitting your manuscript to PLOS ONE. After careful consideration, we feel that it has merit but does not fully meet PLOS ONE’s publication criteria as it currently stands. Therefore, we invite you to submit a revised version of the manuscript that addresses the points raised during the review process.

We look forward to receiving your revised manuscript.

Kind regards,

César Leal-Costa, Ph. D

Academic Editor

PLOS ONE

**Journal requirements:**

“In addition to the named authors, The Global Adolescent and Young Adult Cancer Accord End-of-Life Study Group also included Afaf Girgis, Ahmed Al-Awamer, Anne Kirchhoff, Douglas Fair, Joan Haase, Karen Wernli, Leigh Donovan, Maria Cable, Pamela Mosher, Richard Cohn, Ruwanthie (Amanda) Fernando, and Susan Trethewie. This work was supported by the Inaugural AYA Psycho-Oncology Research Acceleration Grant (2018-2020) funded by the Adolescent and Young Adult Cancer Global Accord, an international alliance comprised of Canteen Australia, Teen Cancer America, and the Teenage Cancer Trust. Ursula Sansom-Daly wishes to thank the AYA Cancer Global Accord mentors who provided helpful guidance in the development and undertaking of this study, in particular A/Prof Pandora Patterson and Dr Fiona McDonald (Canteen Australia), Prof Dan Stark (University of Leeds, UK) and Dr Norma D’Agostino (Princess Margaret Cancer Centre, Canada), and Prof Bradley Zebrack (University of Michigan, USA). Ursula Sansom-Daly is supported by an Early Career Fellowship from the Cancer Institute of New South Wales (ID: 2020/ECF1163) and an Early Career Fellowship from the National Health and Medical Research Council of Australia (APP1111800). Lori Wiener is supported, in part, by the Intramural Program of the National Cancer Institutes, Center for Cancer Research. Claire Wakefield is supported by a Career Development Fellowship from the National Health and Medical Research Council of Australia (APP1143767). The Behavioural Sciences Unit is proudly supported by the Kids with Cancer Foundation, by the Kids Cancer Alliance, as well as a Cancer Council New South Wales Program Grant (PG16-02) with the support of the Estate of the Late Harry McPaul. Meaghann Weaver participated in this project in a private capacity; the views expressed in this article are those of the authors and do not necessarily reflect the position or policy of the U.S. Department of Veterans Affairs, the U.S. Government, or the VA National Center for Ethics in Health Care.”

“In addition to the named authors, The Global Adolescent and Young Adult Cancer Accord End-of-Life Study Group also included Afaf Girgis, Ahmed Al-Awamer, Anne Kirchhoff, Douglas Fair, Joan Haase, Karen Wernli, Leigh Donovan, Maria Cable, Pamela Mosher, Richard Cohn, Ruwanthie (Amanda) Fernando, and Susan Trethewie. This work was supported by the Inaugural AYA Psycho-Oncology Research Acceleration Grant (2018-2020) funded by the Adolescent and Young Adult Cancer Global Accord, an international alliance comprised of Canteen Australia, Teen Cancer America, and the Teenage Cancer Trust. Ursula Sansom-Daly wishes to thank the AYA Cancer Global Accord mentors who provided helpful guidance in the development and undertaking of this study, in particular A/Prof Pandora Patterson and Dr Fiona McDonald (Canteen Australia), Prof Dan Stark (University of Leeds, UK) and Dr Norma D’Agostino (Princess Margaret Cancer Centre, Canada), and Prof Bradley Zebrack (University of Michigan, USA). Ursula Sansom-Daly is supported by an Early Career Fellowship from the Cancer Institute of New South Wales (ID: 2020/ECF1163) and an Early Career Fellowship from the National Health and Medical Research Council of Australia (APP1111800). Lori Wiener is supported, in part, by the Intramural Program of the National Cancer Institutes, Center for Cancer Research. Claire Wakefield is supported by a Career Development Fellowship from the National Health and Medical Research Council of Australia (APP1143767). The Behavioural Sciences Unit is proudly supported by the Kids with Cancer Foundation, by the Kids Cancer Alliance, as well as a Cancer Council New South Wales Program Grant (PG16-02) with the support of the Estate of the Late Harry McPaul. Meaghann Weaver participated in this project in a private capacity; the views expressed in this article are those of the authors and do not necessarily reflect the position or policy of the U.S. Department of Veterans Affairs, the U.S. Government, or the VA National Center for Ethics in Health Care.”

“USD received an AYA Global Accord Psycho-Oncology Research Acceleration Grant numbered RG180972 from the Adolescent and Young Adult Cancer Global Accord (a partnership between Teen Cancer America [https://teencanceramerica.org/], Teenage Cancer Trust [https://www.teenagecancertrust.org/] and Canteen [https://www.canteen.org.au/]). The funders will not have a role in study design, data collection and analysis, decision to publish or preparation of the manuscript.”

4. One of the noted authors is a group or consortium [Afaf Girgis, Ahmed Al-Awamer, Anne Kirchhoff, Douglas Fair, Joan Haase, Karen Wernli, Leigh Donovan, Maria Cable, Pamela Mosher, Richard Cohn, Ruwanthie (Amanda) Fernando, and Susan Trethewie]. In addition to naming the author group, please list the individual authors and affiliations within this group in the acknowledgments section of your manuscript. Please also indicate clearly a lead author for this group along with a contact email address.

Reviewers' comments:

Reviewer's Responses to Questions

**Comments to the Author**

1. Does the manuscript provide a valid rationale for the proposed study, with clearly identified and justified research questions?

Reviewer #1: Partly

Reviewer #2: Yes

2. Is the protocol technically sound and planned in a manner that will lead to a meaningful outcome and allow testing the stated hypotheses?

Reviewer #1: Partly

Reviewer #2: Yes

3. Is the methodology feasible and described in sufficient detail to allow the work to be replicable?

Reviewer #1: Yes

Reviewer #2: Yes

4. Have the authors described where all data underlying the findings will be made available when the study is complete?

Reviewer #1: No

Reviewer #2: No

5. Is the manuscript presented in an intelligible fashion and written in standard English?

Reviewer #1: Yes

Reviewer #2: Yes

6. Review Comments to the Author

You may also provide optional suggestions and comments to authors that they might find helpful in planning their study.

Reviewer #1: Thank you for the opportunity to review the manuscript titled “Thinking globally to improve care locally: A Delphi study protocol to achieve international clinical consensus on best-practice end-of-life communication with adolescents and young adults with cancer”.

This paper is a protocol manuscript that describes the methodology for the planned Delphi study for the adolescent and young adult（AYA）with cancer to develop evidence-based training to support health professionals in facilitating best-practice end-of-life communication in westernized countries.

The study provides an important contribution to world-wide topic about end-of-life communication, especially in AYAs with cancer, however, it needs few major clarifications and some edits to improve.

Major comments

・The definition of end-of-life communication is ambiguous. At least the content (pathology or fatality or prognosis, etc.) for this survey should be clearly explained.

・The authors plan a Delphi study for four questions. However, it is difficult to understand that the four questions are unique to the content of AYAs. Please add and explicitly state. And the authors should add an explanation as to what kind of training this research will lead to for whom.

・As an important point, please describe which database this study will be managed by and how it is stored.

Minor comments

Introduction

・p6, line 112-114: Regarding the description of end-of-life communication, please clearly state who is talking to whom and what. Please specify the need to focus on AYAs in comparison with the previous studies (Pia von Blanckenburg et al. BMJ Open. 2022; Nagelschmidt K et al. BMJ Support Palliat Care. 2021).

・p6, line 115-117: For AYA cancer, the time required for end-of-life communication should vary depending on each pathology, such as leukemia, lymphoma, testicular cancer, and thyroid cancer. This should be closely related to the first question the authors ask. The authors should clearly state the planned study on AYA-specific issues.

・p9, line 194-195: Please clearly state which occupations are specifically referred to as health professionals.

Methods

・p10, line 208-211: It is unclear how many medical doctors, nurses, psychotherapists, etc. are included in the expert team, so please state clearly. If the results are expected to have some impact outside of a particular discipline, it is better to incorporate a various clinical perspective.

Aims

・p11, line 234: Please clearly state the definition of end-of-life communication.

・p11, line 235-238: It is ambiguous how these two "training" questions contribute to solving the p5(line 92-94) and p6 (line 126) problems mentioned in the introduction. There is a risk of getting a snap answer due to an ambiguous question. Please explain in a more understandable manner to improve the Delphi method's validity.

Design

・p13, Table1: Please indicate if the Email counts include conversations between two or all of the participants.

・p14, line 307-310: Please describe how to follow the psychological care including PTSD flashbacks of patients and their families participating through this study.

・p15, line 313-315: Religious background are strongly expected to be largely reflected in end-of-life communication. Please describe where such considerations are taken into account.

・p15, line 328: The authors defined eligible ‘experts’ as health professionals who have provided clinical care to at least five AYAs with cancer who subsequently died. However, does the experience differ depending on whether the member is doing it directly or indirectly? Isn't the number of years of experience more appropriate?

Analysis

・p18, line 36: Generally, the Delphi study aims at consensus building, but in this study, it is described as "consensus will not be defined as reached according to any specific cut-off point". Please clarify why the author does not predefine consensus in this study. The rigorous use of the Delphi technique is required as described in the author’s reference (Jünger S, et al. 2017).

Discussion

・p21, line 442: Please clearly state which facility the study is under the permission of the institutional review board.

Reviewer #2: This is a much needed delphi study protocol. The authors address a real gap in the palliative community for evidence-based guidelines and consensus towards best-practices in EOL communication with the AYA cancer patient population.

Investigator team is multidisciplinary with wide ranging expertise. Would suggest Table 2 to further expound on the various categories of allied health professionals for more clarity (e.g. clinical psychologists? social workers? play therapists?)

Would also suggest AYA (consumer) input to be further broadened to incorporate diverse perspectives. (younger AYA survivors as I note the current age range to be 20-35 yo, parents of AYA survivors, bereaved parents of AYA decedents). Agree that consumer input is imperative in ensuring real-world implementation.

The question of "What is the optimal timing for end-of-life communication with AYAs" is likely to draw varied opinions from the Delphi group especially with prognostic uncertainty within the AYA cancer population. Are the authors providing categorical fields for answers (e.g. "< 1 year prognosis, 3-6 months prognosis, etc) and/or providing free text fields for responses?

- in addition, if Questionnaire 1 and 2 has already been developed, will the authors be sharing within the supplementary file?

Limitations have been well addressed and agree that representative perspectives and guidelines on AYA EOL Communication across AYA Global Accord countries would be extremely useful before broadening to incorporate sociocultural perspectives from other countries.

7. PLOS authors have the option to publish the peer review history of their article (what does this mean?). If published, this will include your full peer review and any attached files.

Reviewer #1: No

Reviewer #2: No

---

## [Author Response · Author response to Decision Letter 0]

13 Jun 2022

1. We have amended the article and figure according to PLOS ONE’s style requirements.

2. Please see below our updated funding statement, with edits bolded and highlighted below.

Funding statement:

“Ursula Sansom-Daly received an AYA Global Accord Psycho-Oncology Research Acceleration Grant numbered RG180972 from the Adolescent and Young Adult Cancer Global Accord (a partnership between Teen Cancer America [https://teencanceramerica.org/], Teenage Cancer Trust [https://www.teenagecancertrust.org/] and Canteen [https://www.canteen.org.au/]). The funders will not have a role in study design, data collection and analysis, decision to publish or preparation of the manuscript. Ursula Sansom-Daly is also supported by an Early Career Fellowship from the Cancer Institute of New South Wales (ID: 2020/ECF1163) and an Early Career Fellowship from the National Health and Medical Research Council of Australia (APP2008300). Lori Wiener is supported, in part, by the Intramural Program of the National Cancer Institutes, Center for Cancer Research. Claire Wakefield is supported by a Career Development Fellowship from the National Health and Medical Research Council of Australia (APP1143767). The Behavioural Sciences Unit is proudly supported by the Kids with Cancer Foundation, by the Kids Cancer Alliance, as well as a Cancer Council New South Wales Program Grant (PG16-02) with the support of the Estate of the Late Harry McPaul. There was no additional external funding received for this study.”

3. Please see below our amended acknowledgements statement, with funding information removed.

Acknowledgements statement: 

“In addition to the named authors, The Global Adolescent and Young Adult Cancer Accord End-of-Life Study Group also included:

Abby Rosenberg; Palliative Care and Resilience Lab, Seattle Children’s Research Institute, Seattle, Washington, United States of America; Cancer and Blood Disorders Center, Seattle Children’s Hospital, Seattle, Washington, United States of America

Afaf Girgis; Centre for Oncology Education and Research Translation (CONCERT), Ingham Institute for Applied Medical Research, UNSW Sydney, Liverpool, New South Wales, Australia.

Ahmed Al-Awamer; Department of Family and Community Medicine, University of Toronto, Toronto, Ontario, Canada; Princess Margaret Cancer Center, Toronto, Ontario, Canada

Anne Kirchhoff; University of Utah School of Medicine, Department of Pediatrics, Division of Hematology/Oncology, Salt Lake City, Utah, United States of America; Huntsman Cancer Institute, Cancer Control & Population Sciences, Salt Lake City, Utah, United States of America

Anne-Sophie Darlington; School of Health Sciences, University of Southampton, Highfield, Southampton, United Kingdom.

Anthony Herbert; Children’s Health Queensland Hospital and Health Service, South Brisbane, Queensland, Australia; Centre for Children’s Research, Queensland University of Technology, Brisbane, Queensland, Australia

Antoinette Anazodo; School of Women’s and Children’s Health, Faculty of Medicine and Health, University of New South Wales Sydney, Kensington, New South Wales, Australia;

Sydney Youth Cancer Service, Prince of Wales/Sydney Children’s Hospital, Randwick, New South Wales, Australia

Celeste Phillips; School of Nursing, Indiana University, Bloomington, Indiana, United States of America.

Claire Wakefield; Behavioural Sciences Unit proudly supported by the Kids with Cancer Foundation, Kids Cancer Centre, Sydney Children’s Hospital, Randwick, New South Wales, Australia; School of Women’s and Children’s Health, Faculty of Medicine and Health, University of New South Wales Sydney, Kensington, New South Wales, Australia.

Douglas Fair; University of Utah School of Medicine Department of Pediatrics, Division of Pediatric Hematology/Oncology, Salt Lake City, Utah, United States of America.

Fiona Schulte; Division of Psychosocial Oncology, Department of Oncology, Cumming School of Medicine, University of Calgary, Calgary, Alberta, Canada; Hematology, Oncology and Transplant Program, Alberta Children’s Hospital, Calgary, Alberta, Canada.

Hanneke Poort; Department of Psychosocial Oncology and Palliative Care, Dana-Farber Cancer Institute, Boston, Massachusetts, United States of America.

Holly Evans; Behavioural Sciences Unit proudly supported by the Kids with Cancer Foundation, Kids Cancer Centre, Sydney Children’s Hospital, Randwick, New South Wales, Australia; School of Women’s and Children’s Health, Faculty of Medicine and Health, University of New South Wales Sydney, Kensington, New South Wales, Australia.

Jennifer Mack; Population Sciences for Pediatric Hematology/Oncology, Dana-Farber Cancer Institute, Boston, Massachusetts, United States of America.

Joan Haase; School of Nursing, Indiana University, Bloomington, Indiana, United States of America; The RESPECT Signature Center at IUPUI, Indianapolis, Indiana, United States of America.

Karen Wernli; Kaiser Permanente Washington Health Research Institute, Oakland, California, United States of America

Leigh Donovan; Collaboraide, Queensland, Australia.

Lori Wiener; Psychosocial Support and Research Program, Pediatric Oncology Branch, Center for Cancer Research, National Cancer Institute, National Institutes of Health, Bethesda, Maryland, United States of America.

Louise Sue; Adolescent and Young Adult Cancer Services Team, Canterbury District Health Board, Christchurch, New Zealand.

Maria Cable; School of Nursing, Midwifery and Health, Coventry University, Coventry, United Kingdom.

Meaghann Weaver; Divisions of Palliative Care & Pediatric Hematology/Oncology, University of Nebraska Medical Center, Lincoln, Nebraska, United States of America; National Center for Ethics in Health Care, Washington DC, Washington, United States of America

Pamela Mosher; Hospital for Sick Children, Toronto, Ontario, Canada.

Richard Cohn; Behavioural Sciences Unit proudly supported by the Kids with Cancer Foundation, Kids Cancer Centre, Sydney Children’s Hospital, Randwick, New South Wales, Australia; School of Women’s and Children’s Health, Faculty of Medicine and Health, University of New South Wales Sydney, Kensington, New South Wales, Australia; 

Ruwanthie (Amanda) Fernando; Palliative Care Service, Liverpool Cancer Therapy Centre, Liverpool Hospital, Liverpool, New South Wales, Australia.

Susan Trethewie; Sydney Children’s Hospital, Randwick, New South Wales, Australia.

Toni Lindsay; Chris O’Brien Lifehouse Cancer Centre, Camperdown, New South Wales, Australia.

Ursula Sansom-Daly; Behavioural Sciences Unit proudly supported by the Kids with Cancer Foundation, Kids Cancer Centre, Sydney Children’s Hospital, Randwick, New South Wales, Australia; School of Women’s and Children’s Health, Faculty of Medicine and Health, University of New South Wales Sydney, Kensington, New South Wales, Australia. The lead author for this group is Ursula Sansom-Daly, contactable on ursula@unsw.edu.au. Ursula Sansom-Daly wishes to thank the AYA Cancer Global Accord mentors who provided helpful guidance in the development and undertaking of this study, in particular A/Prof Pandora Patterson and Dr Fiona McDonald (Canteen Australia), Prof Dan Stark (University of Leeds, UK) and Dr Norma D’Agostino (Princess Margaret Cancer Centre, Canada), and Prof Bradley Zebrack (University of Michigan, USA). Meaghann Weaver participated in this project in a private capacity; the views expressed in this article are those of the authors and do not necessarily reflect the position or policy of the U.S. Department of Veterans Affairs, the U.S. Government, or the VA National Center for Ethics in Health Care.”

4. We have now listed all members of the authorship group in the acknowledgements section. The lead author is now also identified in acknowledgements. 

5. We have also updated our Methods section to include a full ethics statement, see Line 361. 

“The study has been approved by the South Eastern Sydney Local Health District Human Ethics Committee, and online written consent will be obtained.”

 Reviewer’s comments:

1. Have the authors described where all data underlying the findings will be made available when the study is complete?

We will be able to make relevant data available via UNSWorks (https://unsworks.unsw.edu.au) when the study is complete, however, no data has yet been generated as part of the present protocol study.

2. Reviewer 1

a. The definition of end-of-life communication is ambiguous. At least the content (pathology or fatality or prognosis, etc.) for this survey should be clearly explained.

We have made our definition of end-of-life communication more concrete where it is brought up in the Introduction, Line 125. 

“We define end-of-life communication as conversations about death and dying, preferences for medical and psychosocial care towards end of life, including issues of prognosis when cure is not likely, as well as broader quality of life, social, psychological and existential concerns patients may have in the context of a life-limiting illness.[22] These conversations can be between the dying young person and their treating healthcare professionals, families and friends. These conversations often occur in the last 12 months of an individual’s life and/or when a clinical team “would not be surprised if the patient died in the next 12 months”.[23]”

b. The authors plan a Delphi study for four questions. However, it is difficult to understand that the four questions are unique to the content of AYAs. Please add and explicitly state. 

We have added to our explanation of the unique difficulties facing AYAs at end-of-life which justify our focus on AYAs through our aims. See Line 109-112. 

“The fact that these futures may be prematurely cut short is unique to the context of AYAs, and thus highly distressing and developmentally challenging.[9, 10] Further, AYA cancer mortality remains, on average, higher than pediatric cancer mortality.[11-13]. Given these unique challenges, ensuring best-practice end-of-life care and communication for AYAs remains a clinical imperative in this era of novel therapies, precision medicines, and uncertain prognoses.[14]”

We have also explicitly stated at Line 236 that these unique difficulties justify the Aims presented and focus on AYAs. 

“Given the unique challenges associated with AYAs communication around end-of-life, this study will focus on the AYA years. Our study builds on the research on evidence-based standards of care and barriers to implementing the standards established by members of our team.[25, 26, 41]”

c. And the authors should add an explanation as to what kind of training this research will lead to for whom.

We thank the reviewer for enquiring as to the kind of training that will be developed. As this study aims to determine the kind of training which is required by the global AYA cancer workforce, we are unable to firmly predict what kind of training our research will lead to. We have however speculated on elements which may be included in the development of the training and the training itself starting at line 523.

“Building on all the preceding training content development work, as well as the initial intervention mapping[60] exercise to identify potential existing similar training resources and avenues for training dissemination, we will develop a multi-pronged training implementation strategy to expand access to our new training model. We anticipate that this strategy will be guided by the Theoretical Domains Framework,[61, 62] to account for factors influencing behavior change at individual, team and healthcare system levels, together with the Consolidated Framework for Implementation Research,[63] to enable further examination of barriers and enablers at different levels of healthcare organizations.[64] This implementation strategy development will be led through our team, in conjunction with AYA Cancer Global Accord leaders, and other stakeholders with expertise in end-of-life communication education. We anticipate this implementation plan will involve multiple avenues for implementing this training, and will likely include train-the-trainer components.[49, 65] We will look to integrate our new training resources where possible with existing palliative care/end-of-life communication training programs to leverage international engagement and optimize the reach and accessibility of end-of-life communication training to as diverse an international health-professional sample as possible.”

d. As an important point, please describe which database this study will be managed by and how it is stored.

We have updated the Analysis section to include information about how study data will be managed and stored, see Line 384

“Data will be collected using Qualtrics, and then managed with IBM SPSS Statistics Version 27. Data will be stored in files on secure university servers.”

e. Introduction

i. p6, line 112-114: Regarding the description of end-of-life communication, please clearly state who is talking to whom and what. Please specify the need to focus on AYAs in comparison with the previous studies (Pia von Blanckenburg et al. BMJ Open. 2022; Nagelschmidt K et al. BMJ Support Palliat Care. 2021).

We have added to our definition of end-of-life communication information about who is talking to whom, in addition to our explanation of the topics covered, see Line 125. With regards to the need for focus on AYAs, recent research highlighted by Reviewer 1 has looked at barriers to end-of-life communication at the family and young person level, however there is a lack of systematic study of barriers at the healthcare professional level. Further, given the unique system-level settings AYA practitioners work in, there is a need to understand how this care can best be delivered and these barriers addressed via understanding what training is needed to better equip the AYA workforce across different settings.

We have also added citations to the two recent papers suggested by Reviewer 1 at Line 185.

“Numerous barriers may contribute to this gap, at individual, healthcare-team, and hospital-system levels.[22] Recent work has highlighted barriers at a young person and familial level.[44,45] At the individual health professional level, clinicians such as doctors, nurses and allied health professionals can experience considerable distress and counter-transference when health-professionals (who may be relatively young themselves) treat and support dying young people.[45, 46]”

ii. p6, line 115-117: For AYA cancer, the time required for end-of-life communication should vary depending on each pathology, such as leukemia, lymphoma, testicular cancer, and thyroid cancer. This should be closely related to the first question the authors ask. The authors should clearly state the planned study on AYA-specific issues.

We thank the reviewer for their recognition of this important issue, a similar comment was raised by the second reviewer, that timing of end-of-life communication will be dependent on individual disease pathology and stage of the illness. We have added text at Line 316 to describe how we will be addressing this issue.

“Questions regarding optimal timing of end-of-life conversations will attempt to take into account various prognostic stages and illness scenarios. For example, participants will be asked when a particular end-of-life topic is appropriate to introduce to AYAs facing different illness scenarios.”

iii. p9, line 194-195: Please clearly state which occupations are specifically referred to as health professionals.

We have updated the language at Line 185 to include examples of the occupations referred to as health professionals.

“Numerous barriers may contribute to this gap, at individual, healthcare-team, and hospital-system levels.[22] At the individual level, clinicians such as doctors, nurses and allied health professionals can experience considerable distress and counter-transference when health-professionals (who may be relatively young themselves) treat and support dying young people.[44, 45]”

f. Methods

i. p10, line 208-211: It is unclear how many medical doctors, nurses, psychotherapists, etc. are included in the expert team, so please state clearly. If the results are expected to have some impact outside of a particular discipline, it is better to incorporate a various clinical perspective.

We agree that it is important to incorporate various clinical perspectives. However, we apologise for any confusion caused; the list of disciplines included at Line 224 in the revised manuscript are all of the disciplines covered by our team. We have adjusted the language here to try to reduce confusion. 

“Our team is highly multidisciplinary, and includes experts trained in the following areas; psychology/clinical psychology (n=7), social work (n=2), oncology/hematology (n=5), nursing (n=5), palliative care (n=7), public health and cancer epidemiology (n=3), psychiatry (n=1), pediatrics (n=1), education (n=1), and clinical ethics (n=3).”

g. Aims

i. p11, line 234: Please clearly state the definition of end-of-life communication.

As mentioned above, we have explicated our definition of end-of-life communication where it is brought up in the Introduction, Line 125.

ii. p11, line 235-238: It is ambiguous how these two "training" questions contribute to solving the p5(line 92-94) and p6 (line 126) problems mentioned in the introduction. There is a risk of getting a snap answer due to an ambiguous question. Please explain in a more understandable manner to improve the Delphi method's validity.

Thank you for raising this issue. It is not our intention to use the Aims posed as questions as written in our survey instrument, the survey items presented to participants will not be worded as such and will thus be less ambiguous. Methodologies such as ranking questions and value weighting will be used. We have highlighted this in our Aims section at Line 252.

“We aim to answer the following research questions using survey methodologies such as ranking and value weighting:”

h. Design

i. p13, Table1: Please indicate if the Email counts include conversations between two or all of the participants.

We have updated the table to indicate that email counts were between two or more participants, Line 304.

“Table 1. Counts of group-level emails exchanged between two or more investigators by project topics as at June 2021.”

ii. p14, line 307-310: Please describe how to follow the psychological care including PTSD flashbacks of patients and their families participating through this study.

This is an important issue raised by the reviewer, and we would like to assure the reviewer and PLOS ONE that all standard risk mitigation procedures will be followed. Consumer consultation will be facilitated by the study Principal Investigator (USD) who is a clinical psychologist with significant experience supporting AYAs with cancer, alongside the research coordinator (HEE) who has extensive experience interviewing participants in sensitive topics. At various points via verbal and written information, consumers will be reminded that they can stop at any time and to indicate to us if these activities raise distress. If any distress is experienced the research team can support the consumers to access appropriate supports if they aren't already linked in with same. We also wish to note that no adverse events were experienced by patients and families as part of the initial consultation conducted to design the study.

iii. p15, line 313-315: Religious background are strongly expected to be largely reflected in end-of-life communication. Please describe where such considerations are taken into account.

We have added text at Line 313 to indicate that we will take this issue into consideration in our data collection and subsequent analysis. 

“The questionnaire starts with demographic and workplace questions, including questions about religion.”

iv. p15, line 328: The authors defined eligible ‘experts’ as health professionals who have provided clinical care to at least five AYAs with cancer who subsequently died. However, does the experience differ depending on whether the member is doing it directly or indirectly? Isn't the number of years of experience more appropriate?

Thank you for this suggestion. We have given some thought to this issue and are concerned that using number of years' experience would cause specialists to be over-represented, as they are potentially a group likely to care for more dying AYAs in a shorter time span. As such while it is not a perfect measure we will continue with the definition of experts as having provided care to at least five AYAs who have died of cancer.

i. Analysis

i. p18, line 36: Generally, the Delphi study aims at consensus building, but in this study, it is described as "consensus will not be defined as reached according to any specific cut-off point". Please clarify why the author does not predefine consensus in this study. The rigorous use of the Delphi technique is required as described in the author’s reference (Jünger S, et al. 2017).

Thank you for bringing up this important issue. Following Junger's methods, as this study is primarily explorative in nature, we will document the degree of consensus and variability that naturally emerges among our clinician sample in Round 1. Following Junger’s recommendation across Round 2 we will use the commonly applied threshold of 80% item agreement to determine formal consensus in the final analysis of study results. See Line 392.

“As such, due to the exploratory nature of the study and of previous literature[57], we will document the degree of consensus and also variability that naturally emerges among our clinician sample in Round 1, then across Round 2 we will use the commonly used threshold of 80% item agreement to determine formal consensus in the final analysis of study results.”

j. Discussion

i. p21, line 442: Please clearly state which facility the study is under the permission of the institutional review board.

To clarify this issue, we have stated which university has applied for the IRB permissions, see Line 479.

“Since the commencement of this project, our team via The University of New South Wales has completed three separate ethical (Institutional Review Board) application processes internationally, one governance application, and three protocol amendments for updates to questionnaire content.”

3. Reviewer 2

a. Would suggest Table 2 to further expound on the various categories of allied health professionals for more clarity (e.g. clinical psychologists? social workers? play therapists?)

We have added examples of allied health professionals for clarity in Table 2 (Line 364), thank you for this suggestion. 

“Allied health professionals (e.g. psychologists, social workers, play therapists):”

b. Would also suggest AYA (consumer) input to be further broadened to incorporate diverse perspectives. (younger AYA survivors as I note the current age range to be 20-35 yo, parents of AYA survivors, bereaved parents of AYA decedents). Agree that consumer input is imperative in ensuring real-world implementation.

The age range of 20-35 is the range for the already-recruited 24 consumers who consulted on study design. As part of this project, we will recruit further consumers to consult on results, who will range in age from 15 to 40. With regards to parents of AYAs, we have taken this feedback on board and will be recruiting a parent group which will be run separately to the AYA group (Line 401). 

“We will establish an AYA Advisory Panel comprising an additional 10-15 AYAs aged 15 to 39 to provide feedback on the training priorities identified through our Delphi questionnaire. A further advisory panel of approximately 5 parents of AYA cancer patients/survivors and bereaved parents will be run separately. Following the analysis of results from the Delphi questionnaire Rounds 2 and 3, we will present a written lay summary of key findings to AYAs and parents recruited as consumers. They will be asked to complete an online questionnaire which asks them to reflect on the summary findings with reference to their own cancer-related experiences, record the extent to which the recommendations for health-professional training matches their experience, and identify gaps in the findings. Next, AYAs and parents will be invited to participate in an online focus group, during which the themes of the findings and their reflections on these will be further explored. Finally, they will be invited to share reflections in the publication of the final results. They will be asked to describe ways in which they would like to see AYA and parent input integrated into the development, and potentially the delivery and implementation, of future training programs. This input will not be used to alter the outcomes of the Delphi questionnaire, but rather will contribute a vital layer of qualitative data that will be integral to the interpretation of the data through a consumer-driven lens. The involvement of AYAs and parents as partners in shaping research, particularly AYA health research, is an ethical imperative,[58] and can contribute to ensuring that research needed and valued by AYAs translates well to real-world practice.[59, 60]”

c. The question of "What is the optimal timing for end-of-life communication with AYAs" is likely to draw varied opinions from the Delphi group especially with prognostic uncertainty within the AYA cancer population. Are the authors providing categorical fields for answers (e.g. "< 1 year prognosis, 3-6 months prognosis, etc) and/or providing free text fields for responses?

We have added text at Line 316 to describe how we will be addressing this issue.

“Questions regarding optimal timing of end-of-life conversations will attempt to take into account various prognostic stages and illness scenarios. For example, participants will be asked when a particular end-of-life topic is appropriate to introduce to AYAs in different illness scenarios.”

d. In addition, if Questionnaire 1 and 2 has already been developed, will the authors be sharing within the supplementary file?

We thank the reviewer for this suggestion. Questionnaire 1 has been developed as part of Round 1, and Questionnaire 2 will be developed following data collection for Questionnaire 1. Given that the survey has not yet been disseminated, and this manuscript may be read by future participants in the survey, we would prefer to keep the exact questionnaire items for Q1 restricted to study personnel for the time being. We plan to make the full questionnaires accessible when we publish the study results.

Please address any correspondence to the first author, Dr Ursula Sansom-Daly, Behavioural Sciences Unit, Kids Cancer Centre, Level 1 South Wing, Sydney Children’s Hospital, Randwick, NSW 2031, Sydney, AUSTRALIA. Tel +612 9382-3114, e-mail: ursula@unsw.edu.au. 

Many thanks again for your consideration of this manuscript.

Yours sincerely,

Ursula Sansom-Daly, B.Psych (Hons I), M. Psychol (Clin), PhD.

---

## [Decision Letter · Decision Letter 1]

20 Jun 2022

Thinking globally to improve care locally: A Delphi study protocol to achieve international clinical consensus on best-practice end-of-life communication with adolescents and young adults with cancer

PONE-D-22-03862R1

Dear Dr. Sansom-Daly,

We’re pleased to inform you that your manuscript has been judged scientifically suitable for publication and will be formally accepted for publication once it meets all outstanding technical requirements.

Kind regards,

César Leal-Costa, Ph. D

Academic Editor

PLOS ONE

Additional Editor Comments (optional):

Reviewers' comments:

Reviewer's Responses to Questions

**Comments to the Author**

1. Does the manuscript provide a valid rationale for the proposed study, with clearly identified and justified research questions?

Reviewer #1: Yes

Reviewer #2: Yes

2. Is the protocol technically sound and planned in a manner that will lead to a meaningful outcome and allow testing the stated hypotheses?

Reviewer #1: Yes

Reviewer #2: Yes

3. Is the methodology feasible and described in sufficient detail to allow the work to be replicable?

Reviewer #1: Yes

Reviewer #2: Yes

4. Have the authors described where all data underlying the findings will be made available when the study is complete?

Reviewer #1: Yes

Reviewer #2: Yes

5. Is the manuscript presented in an intelligible fashion and written in standard English?

Reviewer #1: Yes

Reviewer #2: Yes

6. Review Comments to the Author

You may also provide optional suggestions and comments to authors that they might find helpful in planning their study.

Reviewer #1: Thank you for the opportunity to review this paper. My questions have been answered properly and I have no further questions.

Reviewer #2: No further comments as author has adequately addressed previous review comments. I wish them success as they embark on this study.

7. PLOS authors have the option to publish the peer review history of their article (what does this mean?). If published, this will include your full peer review and any attached files.

Reviewer #1: No

Reviewer #2: No

---

## [Editor Report · Acceptance letter]

29 Jun 2022

PONE-D-22-03862R1 

Thinking globally to improve care locally: A Delphi study protocol to achieve international clinical consensus on best-practice end-of-life communication with adolescents and young adults with cancer 

Dear Dr. Sansom-Daly:

I'm pleased to inform you that your manuscript has been deemed suitable for publication in PLOS ONE. Congratulations! Your manuscript is now with our production department. 

Kind regards, 

on behalf of

Dr. César Leal-Costa 

Academic Editor

PLOS ONE